# Identification and Validation of STC1 Act as a Biomarker for High-Altitude Diseases and Its Pan-Cancer Analysis

**DOI:** 10.3390/ijms25169085

**Published:** 2024-08-21

**Authors:** Qiong Li, Zhichao Xu, Qianhui Gong, Xiaobing Shen

**Affiliations:** 1Key Laboratory of Environmental Medicine Engineering, Ministry of Education, School of Public Health, Southeast University, Nanjing 210003, China; joan_tiny@163.com (Q.L.); xuzhichao_24@163.com (Z.X.); gongqh2717@163.com (Q.G.); 2Department of Epidemiology and Health Statistics, School of Public Health, Southeast University, Nanjing 210003, China

**Keywords:** STC1, high-altitude disease, pan-cancer, tumor microenvironment, immune infiltration

## Abstract

High-altitude diseases, including acute mountain sickness (AMS), high-altitude cerebral edema (HACE), and high-altitude pulmonary edema (HAPE), are closely related to an individual’s ability to adapt to hypoxic environments. However, specific research in this field is relatively limited, and further biomarker research and clinical trials are needed to clarify the exact role and potential therapeutic applications of key genes in high-altitude diseases. This study focuses on the role of the STC1 gene in high-altitude diseases and explores its expression patterns in different types of cancer. By using gene expression data analysis and functional experiments, we identified STC1 as a key gene affecting the development of altitude sickness. In addition, we also conducted expression and mutation analysis on STC1 in various cancer samples and found significant differences in the expression of this gene in 13 types of malignant tumors, which is associated with the hypoxic state in the tumor microenvironment. In addition, STC1 is significantly associated with patient prognosis and influences tumor immunity by mediating six types of immune cells (CD8+T cells, CD4+T cells, neutrophils, macrophages, monocytes, and B cells) in the tumor microenvironment. The expression and diagnostic value of STC1 were confirmed through GEO datasets and qPCR testing, indicating consistency with the results of bioinformatics analysis. These results indicate that STC1 is not only an important factor in the adaptive response to high-altitude diseases but may also play a role in the adaptation of cancer to low-oxygen environments. Our research provides a new perspective and potential targets for the discovery of biomarkers for high-altitude diseases and cancer treatment.

## 1. Introduction

Acclimatization in high-altitude environments is a complex and crucial process. However, at altitudes exceeding 2500 m, the reduced oxygen content can cause the rapid onset of high-altitude diseases in individuals from lowland areas who have not adapted [1]. High-altitude illnesses, such as acute mountain sickness (AMS), high-altitude cerebral edema (HACE), and high-altitude pulmonary edema (HAPE), are common physiological responses to the low oxygen conditions at high altitudes. These illnesses lead to internal imbalances in several physiological systems, including the brain, lungs, and immune system. The regulatory role of the immune system is especially critical [2]. In a low-oxygen environment, the body must first detect fluctuations in oxygen levels. This is primarily accomplished via distinct cellular and molecular mechanisms, such as the activation of HIF (hypoxia-inducible factor). HIF facilitates the onset of several biological processes, such as the elevated production of hemoglobin and the creation of new blood vessels, processes essential for the proper functioning of immune cells [3]. HIF also plays a role in modulating the functions of immune cells, including macrophages and T cells, to enhance their adaptability to hypoxic conditions [4]. In addition, high-altitude environments may also trigger inflammatory reactions in the body. On the one hand, hypoxia can act as a stress signal, inducing the release of inflammatory factors such as IL-6 and TNF-α, which helps promote tissue repair and protect the body from further damage [5]. On the other hand, the immune system also needs to regulate these reactions to prevent tissue damage or other health problems caused by excessive inflammation, such as high-altitude pulmonary edema. Individuals living in high-altitude areas over an extended period demonstrate adaptive immune adjustments, potentially encompassing alterations in T and B cell activities [6]. Research indicates that certain aspects of adaptive immunity differ in individuals residing at high altitudes. For instance, antibody production might be reduced, while the response of certain memory T cells might be heightened, thereby bolstering resistance to environmental pressures [7]. Therefore, genetic research on high-altitude diseases, especially those involving the immune system, hemoglobin concentration, cardiopulmonary function, and oxygen transport efficiency, can indeed provide important information in understanding the mechanisms of human adaptation to high-altitude environments.

Although high-altitude disease and cancer represent two distinct pathological conditions, they could potentially share key biomarkers and signaling pathways at the molecular level. For example, hypoxia is a key characteristic of high-altitude disease and a common phenomenon in the microenvironment of many types of cancer. Tumor hypoxia is a key factor in cancer development and treatment resistance. Performing pan-cancer analysis on the key genes related to high-altitude diseases, that is, studying the expression and mutation of these genes across various types of cancer, may hold unique value in certain situations. If these genes play a role in regulating cell responses to hypoxic environments, they may also be associated with hypoxia in the tumor microenvironment and have potential importance in cancer research. Therefore, we hypothesize that key immune genes linked to high-altitude diseases could also be relevant to the hypoxic immune microenvironment of tumors. If these genes exhibit abnormal expression or functional mutations across various types of cancer, investigating their role within the pan-cancer context may not only reveal their multifunctionality but also aid in the development of novel cancer treatment strategies, particularly for cancers that are challenging to treat due to hypoxia.

## 2. Results

### 2.1. Identification of the DE-miRNAs and DE-mRNAs

After data preprocessing, a total of 21 DE-miRNAs (6 up-regulated and 15 down-regulated) were identified between the 13 AMS and 9 non-AMS groups (Appendix A). Figure 1a,b illustrates the box diagram for sample quality control and a volcano plot of the DE-miRNAs. In addition, analysis of the GSE52209 dataset revealed 200 DE-mRNAs (173 up-regulated and 27 down-regulated) (Appendix A), which are depicted in Figure 1c,d. A total of 817 target genes were identified according to the DE-miRNAs of GSE90500 (Appendix A).

### 2.2. Identification of the Key Genes

Common DE-mRNAs were identified as the overlapping genes between the 817 target genes of DE-miRNAs identified in GSE90500 and the 200 DE-mRNAs identified in GSE52209. In addition, STC1 was identified as the overlapping gene between the 1793 immune-related genes (Appendix A) and 11 common DE-mRNAs (Figure 2e).

### 2.3. PPI Network and Function Enrichment of STC1

To study the function of STC1, we used the STRING database to explore STC1-binding proteins and GO terms and conducted functional enrichment analysis for STC1-related genes using GSEA. As shown in Figure 2a, we obtained 10 STC1-binding proteins, including GCG, FFAR4, TNF, TNFAIP6, MBTPS1, PAPPA, STC2, SLC34A1, PYY, and CCK. We used cytoscape (version 3.9.1) software to construct a network of STC1-binding proteins and GO terms (Figure 2b). Subsequently, we conducted the GO enrichment analysis of 10 targeted binding proteins, revealing that the primary biological process (BP) comprised cell communication, cellular response to stimulus, signal transduction, regulation of localization, regulation of transport, regulation of ion transport, response to hormone, feeding behavior, regulation of anion transport, and response to vitamin. The molecular function (MF) was mainly enriched in transferase activity, transferring one-carbon groups, methyltransferase activity receptor ligand activity, and hormone activity (Figure 2c).

GSEA was carried out to evaluate the biological importance of STC1 expression in a variety of malignancies. The focus of this analysis was on the signaling pathways linked to STC1. Figure 2d–g display some of the findings of an examination of the KEGG pathways as well as an analysis of the HALLMARK keywords. According to the KEGG enrichment term, high expression of STC1 is primarily associated with the glycosaminoglycan biosynthesis pathway, the cell cycle, and DNA replication. On the other hand, low expression of STC1 is primarily associated with the metabolism of xenobiotics by Cytochrome P450, taurine, and hypotaurine metabolism, linoleic acid metabolism, and arachidonic acid metabolism. According to the findings of the GSEA on HALLMARK terms, high expression of STC1 is associated with multiple pathways, including hypoxia, glycolysis and gluconeogenesis, and mTORC1 signaling. On the other hand, low expression of STC1 is associated with the peroxisome, fatty acid metabolism, KRAS signaling, and bile acid metabolism.

### 2.4. Analysis of STC1 Expression between Normal and Tumor Samples

We investigated the physiologic STC1 protein expression levels in different cell lines using the HPA database and discovered that it was detected in many cell lines and had low cancer specificity (Figure 3a). The expression level of STC1 was higher in sarcoma, brain cancer, thyroid cancer, pancreatic cancer, and kidney cancer. Then, relative plasma concentrations of proteins in the blood of patients with different types of cancer were measured by proximity extension assays (PEAs). We found that STC1 is up-regulated in chronic lymphocytic leukemia, acute myeloid leukemia, myeloma, diffuse large B-cell lymphoma, ovarian cancer, and lung cancer (Figure 3b). STC1 expression in the majority of cancer tissues showed weak to moderate cytoplasmic positivity with distinct extracellular staining (Figure 3c). Meanwhile, STC1 protein expression was elevated, as determined by immunohistochemistry (IHC), in liver hepatocellular carcinoma (LIHC), pancreatic adenocarcinoma (PAAD), colon adenocarcinoma (COAD), and stomach adenocarcinoma (STAD) compared to normal tissues, while it had low expression in several cancers, such as thyroid carcinoma (THCA), kidney chromophobe (KICH), and kidney renal papillary cell carcinoma (KIRP) (Figure 3d). As shown in Figure 3e, the TIMER database found that STC1 expression varied significantly between tumor and normal tissue in 13 different forms of cancer. STC1 was highly expressed in cholangiocarcinoma (CHOL), COAD, glioblastoma (GBM), head and neck squamous cell carcinoma (HNSC), liver hepatocellular carcinoma (LIHC), pheochromocytoma and paraganglioma (PCPG), rectum adenocarcinoma (READ), stomach adenocarcinoma (STAD), and uterine corpus endometrial carcinoma (UCEC). In contrast, tumor tissues have lower levels of STC1 compared to healthy tissues within bladder urothelial carcinoma (BLCA), KICH, KIRP, and THCA.

### 2.5. STC1 Gene Acted as a Survival Prognosis Biomarker

By doing a survival analysis, we estimated the correlation between STC1 expression and the outcome of patients who were included in the pan-cancer dataset by using forest plots, including OS (Figure 4a), DPS (Appendix A), and PFI (Appendix A). Kaplan–Meier analysis revealed that a greater level of STC1 expression was significantly connected to a lower OS in uveal melanoma (UVM), BLCA, cervical squamous cell carcinoma and endocervical adenocarcinoma (CESC), head and neck squamous cell carcinoma (HNSC), GBM, pan-kidney cohort, brain lower grade glioma (LGG), LUAD, and stomach and esophageal carcinoma (STES) (Figure 4b–j). In addition, an examination of the DPS data (shown in Appendix A) revealed links between lower STC1 expression and a more favorable outcome in patients with BLCA, CESC, GBM, HNSC, KIRP, LGG, LUAD, and STAD. Regarding the relationship between STC1 expression and PFI, patients with ACC, BLCA, CECS, GBM, HNSC, KIRP, LGG, LUAD, LUSC, and UVM who had low levels of STC1 expression had a longer survival time (Appendix A). These findings made it abundantly evident that STC1 expression was significantly related to patient outcomes across a wide variety of cancer types.

### 2.6. STC1 Expression in Immune or Molecular Subtypes of Cancers and Clinical Phenotypes

We examined the correlations between STC1 differential expression and immune subtypes or molecular subtypes from the TISIDB database. It was indicated that STC1 expression differed in immune subtypes in 30 cancers (Figure 5a), while its expression differed in molecular subtypes in 17 cancers (Figure 5j). Moreover, we visualized the association of STC1 expression with different immune subtypes of eight cancer types, including BLCA, BRCA, COAD, KIRC, LUAD, MESO, OV, and STAD (Figure 5b–i).

We also explored the relationship between STC1 expression and different molecular subtypes in eight cancer types (Figure 5k–r). For BRCA, the highest expression level of STC1 among the molecular subtypes was LumA (Figure 5k). For STAD, the highest expression level of STC1 among the molecular subtypes was HM-SNV (Figure 5l). For PRAD, the highest expression level of STC1 among the molecular subtypes was 7-IDH1 (Figure 5m). For LGG, the highest expression level of STC1 among the molecular subtypes was G-CIMP-low (Figure 5n). For LUSC, the highest expression level of STC1 among the molecular subtypes was basal (Figure 5o). For KIRP, the highest expression level of STC1 among the molecular subtypes was C2c-CIMP (Figure 5p). For COAD, the highest expression level of STC1 among the molecular subtypes was HM-SNV (Figure 5q). For UCEC, the highest expression level of STC1 among the molecular subtypes was MSI (Figure 5r). We observed differential expression of the STC1 gene in samples with different clinical stages in 9 tumors, such as LUAD (*p* < 0.0001), COADREAD (*p* = 0.02), STES (*p* < 0.0001), KIRP (*p* < 0.0001), KIPAN (*p* < 0.0001), STAD (*p* < 0.0001), HNSC (*p* < 0.0001), LUSC (*p* = 0.05), and BLCA (*p* = 0.0001), as shown in Figure 5s and Appendix A.

### 2.7. Immune Cell Infiltration Analysis of STC1 in All Malignancies

Figure 6a displays tumors with the correlation coefficients that were found between the degree of infiltration and STC1 expression for all 6 kinds of immune cells (CD8+ T cells, CD4+ T cells, neutrophils, macrophages, monocytes, and B cells) using several algorithms. We visualized several correlation plots with significant r-values based on different algorithms (Figure 6b–g). CD8+ T cells were negatively related to STC1 in BRCA (Figure 6b). Neutrophils were positively related to STC1 in most cancers, especially COAD and STAD (Figure 6c,d). CD4+ T cells were negatively related to STC1 in THCA (Figure 6e). Macrophages were positively related to STC1 in LGG (Figure 6f). B cells were positively related to STC1 in KIRP (Figure 6g).

### 2.8. Single-Cell Analysis of STC1 in Cancers

To clarify STC1 mRNA expression in different pan-cancer cell types, we obtained single-cell RNA sequencing data for BRCA, CHOL, KIRC, and STAD from GEO datasets for analysis. It was shown that endothelial cells exhibited higher STC1 expression than other cell types in BRCA, CHOL, KIRC, and STAD (Figure 7a–d).

To further study the latent role of STC1 in cancer, we investigated the function of STC1 at the single-cell level using the CancerSEA database (Figure 7e). The results displayed that STC1 was positively linked with EMT, metastasis, cell cycle, and DNA damage of LUAD. In RCC, STC1 had a positive relationship with hypoxia, metastasis, angiogenesis, and differentiation. In melanoma (MEL), STC1 had a positive relationship with inflammation, angiogenesis, EMT, metastasis, quiescence, hypoxia, and apoptosis. STC1 was positively related to hypoxia and metastasis in BRCA. However, there was a negative relationship between STC1 and invasion in OV. STC1 was negatively related to DNA repair and DNA damage in UM. In RB, STC1 was positively related to angiogenesis, differentiation, and inflammation and was negatively related to DNA repair, cell cycle, and DNA damage (Figure 7f). There is no functional state that is significantly related to STC1 in GBM.

### 2.9. The Landscape of the STC1 Mutation Profile in Different Tissues

We looked at how often mutations occurred in STC1 by using the TCGA database (15,230 species within 32 research projects) and the cBioPortal database. Our results demonstrated that ovarian cancer, lung cancer, and ovarian epithelial tumors shared a relatively high alteration frequency of STC1. STC1 was altered in 14.55% of 110 cases (Deep Deletion was 14.55%) in ovarian cancer, 13.16% of 38 cases (Amplification was 5.26%, Deep Deletion was 7.89%) in lung cancer, and 7.36% of 584 cases (Mutation was 0.34%, Amplification was 0.17%, Deep Deletion was 6.85%) in the ovarian epithelial tumor. However, the mutation level with STC1 is low in most cancers (Figure 8a,b). A total of 72 mutation sites were found. Of these, 65 were missense mutations, 3 were truncating mutations, 1 was an inframe mutation, and 3 were splice mutations (Figure 8c).

### 2.10. STC1 Expression in Relation to TMB, MSI, MMR Gene Mutation, and DNA Methylation of STC1

In this work, we investigated whether or not there was a connection between the expression of STC1 and the infiltration scores of common immune cell types in 33 different forms of cancer. The stromal and immune cell scores were calculated with the use of the ESTIMATE algorithm, and the correlations between STC1 expression levels and the stromal and immune cell scores were investigated. According to the findings, a significant correlation existed between the level of STC1 expression and the ESTIMATEScore, ImmuneScore, and StromalScore in GBMLGG, KIPAN, and COADREAD (Figure 9a–c).

We looked at the connections between STC1 and TMB and MSI to discover whether or not there is a connection between the activity of STC1 and mutations in certain forms of cancer. There is a significant (*p* < 0.05) association between the levels of STC1 expression and TMB in ten different forms of cancer. According to the findings, STC1 has a strong positive correlation with TMB in a variety of diseases, including THYM, COAD, LGG, and others (Figure 9d). On the other hand, there is an inverse relationship between the expression of STC1 and TMB in THCA, GBM, KIRP, and other samples (Figure 10d). On the other hand, the connection between STC1 expression and MSI reached statistical significance (*p* < 0.05) in eight different cancer types. There is a positive association between STC1 and MSI in COAD, TGCT, UCEC, SARC, and READ; however, there is a negative correlation between STC1 and MSI in DLBC, HNSC, and BLCA (Figure 9e).

We investigated the link between STC1 expression and many MMR genes to determine whether or not DNA methylation of STC1 directly influences the onset and progression of cancer. According to the findings, there is a substantial correlation between the expression of STC1 and the expression of MMR genes in 21 different forms of cancer (*p* < 0.05), significantly in THCA, PAAD, KIRP, and READ. In the majority of these cancer types, a strong and positive correlation was found between MSH6, MSH2, PMS2, and STC1 (Figure 9f). It is interesting to note that the expression of STC1 in THCA has a strong and positive correlation with each of the five MMR genes, which suggests that MMR regulation may play a role. In addition, we looked at the possibility of a connection between STC1 and the four DNA methyltransferases (DNMT1, DNMT2, DNMT3A, and DNMT3B). In twenty different forms of cancer, including PAAD, THCA, and UVM, STC1 expression is shown to be significantly related to these four DNA methyltransferases (Figure 9g). Based on these findings, it seems that STC1 may influence the growth of tumors in many malignancies by acting as a mediator in the processes of DNA repair and DNA methylation.

### 2.11. Validation of STC1 Expression and Its Diagnostic Value in Pan-Cancer GEO Datasets

Firstly, we demonstrated that the expression level of STC1 was higher in HAPE patients than in control volunteers, which is similar to the expression in GSE52209 (Figure 10a,b). Secondly, we found that the expression level of STC1 was higher in RAW 264.7 cells cultured in hypoxia than in normoxia, and was consistent with the results of GSE145935 (Figure 10c,d). The results are consistent with the results of the bioinformatics analysis. Finally, we obtained eight different cancer-related GEO datasets from the GEO database to confirm the difference in STC1 levels between normal and tumor samples. These findings were entirely in line with our earlier research. The ROC curve was used to assess the pan-cancer diagnostic value of STC1 from these eight GEO datasets. The results showed that STC1 had a certain accuracy in predicting 8 cancer types, including brain tumors (AUC = 0.659, CI: 0.553–0.764) (Figure 11a), breast cancer (AUC = 0.760, CI: 0.662–0.858) (Figure 11b), gastric cancer (AUC = 0.967, CI: 0.938–0.997) (Figure 11c), HCC (AUC = 0.857, CI: 0.713–1.000) (Figure 11d), lung adenocarcinomas (AUC = 0.757, CI: 0.666–0.847) (Figure 11e), ovarian cancer (AUC = 0.920, CI: 0.801–1.000) (Figure 11f), renal cell carcinoma (AUC = 0.768, CI: 0.688–0.847) (Figure 11g), and colorectal cancer (AUC = 0.795, CI: 0.678–0.911) (Figure 11h). These findings suggest that STC1 may serve as a reliable predictor of these cancers.

## 3. Discussion

Despite the important interplay between hypoxia and the immune system, specific molecular mechanisms for adapting to hypoxia in high-altitude illnesses or cancers remain poorly understood. In this study, the STC1 gene was identified as a critical gene linked to high-altitude diseases. Additionally, by conducting a pan-cancer analysis of STC1, the expression patterns and functional roles of STC1 in various cancer types were discerned, shedding light on potential biomarkers or therapeutic targets. To our knowledge, this study provides the first evidence that STC1 is a gene regulated by hypoxia in high-altitude diseases.

Stanniocalcin 1, also known as STC1, is a hormone that regulates low blood calcium levels and is a member of a class of glycoproteins with a molecular weight of 56 kDa. It is produced across a broad spectrum of tissues and likely exerts autocrine or paracrine effects [8]. This protein may play a role in regulating renal and intestinal calcium and phosphate transport, cellular metabolism, and single-cell calcium/phosphate homeostasis, and it influences wound healing, steroid production, cellular metabolism, angiogenesis, carcinogenesis, and inflammation [9]. In non-tumor diseases, STC1 affects the function of immune cells. Previous studies have reported that STC1 is an anti-inflammatory protein in macrophages that inhibits cross-endothelial migration of human macrophages and T lymphocytes [10], reduces macrophage response to chemokines [11], weakens intracellular second messenger signals through calcium, and stabilizes endothelial tight junctions in cytokine-treated endothelial monolayers [12]. Mammalian STC1 reduces the fluidity of macrophages and their response to chemokines. Exogenous STC1 is internalized by macrophages and localized in mitochondria, indicating the role of circulating and/or tissue-derived STC1 in regulating macrophage function [13].

In research on high-altitude diseases and adaptation to high-altitude environments, the STC1 gene has not been reported. However, STC1 plays some important physiological roles in response to hypoxic environments. Under hypoxic conditions, the expression of STC1 can be up-regulated, aiding cells in adapting to hypoxic stress. For example, STC1 can alleviate cellular stress responses under hypoxic conditions, reduce the generation of reactive oxygen species (ROS) through antioxidant effects, and protect cells from damage [14]. STC1 has shown the ability to promote angiogenesis in certain studies, which is particularly important in high-altitude and low-oxygen environments as new blood vessel formation helps improve tissue oxygen supply. STC1 may also be involved in regulating inflammatory responses, helping to control the common excessive inflammatory state in high-altitude environments. In summary, STC1 can serve as a key gene regulating the immune cell function of high-altitude diseases.

Understanding the role of the key gene STC1 in cancer, which is also implicated in high-altitude diseases, can aid doctors and researchers in developing more precise treatment strategies. By examining the roles of key genes associated with high-altitude diseases across various cancers, researchers can achieve a more thorough understanding of STC1’s functions under diverse physiological and pathological conditions. This cross-disease analysis contributes to the creation of a more integrated biological model, potentially unveiling previously unknown pathological mechanisms. Accumulating evidence indicates that STC1 deregulation is linked to various cancers, including breast cancer [15], hepatocellular carcinoma (HCC) [16], glioblastoma multiforme (GBM) [17], pancreatic ductal adenocarcinoma [18], papillary thyroid carcinoma (PTC) [19], ovarian cancer [20], and gastric cancer [21]. Moreover, a recent meta-analysis suggests that elevated STC1 levels could adversely affect the prognosis of patients with solid tumors [22].

In this study, the STC1 gene was strongly expressed in 13 types of cancer but had only low expression in KICH and KIRP. Exploring the reasons for this phenomenon reveals the presence of HIF-1 binding motifs in the STC1 gene promoter and characterizes the related gene transcription activation mechanisms. Therefore, hypoxia-inducible factor (HIF)-1 can activate the STC1 promoter [23]. Meanwhile, HIFs regulate the main transcriptional pathways of the cellular hypoxia response and are negatively regulated by von Hippel Lindau factor (VHL) [24]. VHL is a component of the E3 ligase complex that can promote the degradation of HIF–α [25]. Research has shown that the characteristic of clear cell renal cell carcinoma (ccRCC) is the loss of tumor suppressor VHL function [25]. Therefore, we assume that the loss of VHL function leads to a decrease in the degradation of HIF factors, which inhibit the activation of the STC1 promoter and result in a decrease in the expression of STC1 in KICH and KIRP.

Multiple studies have shown a correlation between overexpression of STC1 and a bad prognosis for patients. This overexpression of STC1 is characterized by high levels of mRNA and protein in serum and malignant tissues when compared to the normal equivalents. It has been shown that a higher STC1 protein level in tumor tissues is related to a worse disease-free survival rate as well as an overall lower survival rate [26]. According to research, STC1 has the potential to operate as a prognostic factor through the HIF-1α/STC1/PI3K-AKT axis in the evolution of PDAC and chemoresistance [18]. According to the findings of another piece of study, STC1 has the potential to serve as a prognostic indicator for individuals afflicted with metastatic ovarian cancer using the FOXC2/ITGB6 signaling axis [20]. Due to clinical evidence, higher STC1 expression in tumor tissues was linked to shorter DFS and OS. STC1 may have a negative connection with prognosis [27]. Moreover, STC1 expression is significantly associated with MMR, TMB, and MSI, as well as DNA methylation, all of which indicate that STC1 may affect cancer prognosis and immunology. This is indicated by the fact that STC1 expression is closely associated with immune infiltration.

During our research, we explored the correlations between STC1 differential expression and pan-cancer immune subtypes and molecular subtypes from the TISIDB database. We concluded that the mRNA expression and protein levels of STC1 in some forms of cancer are connected to the patient’s stage of the disease. Additionally, Li et al. discovered that an elevation in STC1 immunostaining was connected to adverse clinicopathological indicators such as lymph node metastasis, ATA risk score, and TNM stage [28]. These are all indicators of the progression of the patient’s cancer. A significant correction was shown with both tumor diameter and tumor stage when strong STC1 expression was present [22]. We suspected that STC1 may cause cancer by influencing many processes that are linked with tumors, including cell proliferation, survival, migration, and invasion.

Furthermore, our findings demonstrate a strong correlation between STC1 expression and the infiltration levels of immune cells, including CD8+ T cells, CD4+ T cells, neutrophils, macrophages, monocytes, and B cells. This suggests that STC1 may influence the tumor immune response by modulating the activation, migration, or functional dynamics of these immune cell populations. STC1 may influence the infiltration and activity of CD8+ T cells and CD4+ T cells by modulating pro-inflammatory or immunosuppressive factors, such as TGF-β, within the tumor microenvironment [29]. STC1 may indirectly modulate the cytotoxic activity of CD8+ T cells through the regulation of IL-17 expression. STC1 may influence CD4+ T cells to secrete cytokines, including IL-17, IL-31, and IL-33, which modulate immune responses via autocrine or paracrine mechanisms, thereby either amplifying or suppressing immune activity within the tumor microenvironment [30]. Similarly, STC1 may participate in tumor progression by affecting the polarization state of macrophages (M1 anti-tumor or M2 pro tumor). Additionally, STC1 may impact T cell persistence through its anti-apoptotic and pro-survival properties, thereby modulating their functional role within tumors [29].

The microenvironment of the tumor plays a critical role in its development, progression, metastasis, and chemoresistance [31]. The hypoxic situation that is caused by fast proliferation and relatively poor vascularization of the tumor mass is a notable aspect of the microenvironment of the tumor [32]. In response to a hypoxic microenvironment, the expression of hypoxia-inducible factor (HIF) rises in cancerous cells. This, in turn, triggers overexpression of STC1 to enhance cell proliferation and inhibit apoptosis, which in turn promotes metastasis and invasion [31]. A concentration-dependent effect of STC1’s stimulation of mitochondrial electron transport chain activity and calcium transport has led to its classification as a stimulator of mitochondrial respiration [33]. Because of the Warburg effect, STC1 causes a change from mitochondrial respiration to a more glycolytic metabolic profile, hence enhancing tumor cells’ tolerance to hypoxia and preventing them from apoptosizing [34]. Single-cell analysis illustrates that STC1 is predominantly expressed in endothelial cells. In addition to this, STC1 promotes tumor neo-angiogenesis, which is the formation of new blood vessels inside a tumor. This new vascular system makes it possible for cancer cells to obtain sufficient oxygen and nutrients for survival and proliferation, and it also encourages distant metastasis [35].

## 4. Materials and Methods

### 4.1. Data Source and Processing

Two datasets, GSE90500 and GSE52209, were downloaded from the GEO (https://www.ncbi.nlm.nih.gov/geo/ (accessed on 9 June 2022)) database for identifying the key genes of high-altitude diseases. The RNA-seq data GSE90500 contained 13 patients with AMS and 9 non-AMS volunteers. The microarray data GSE52209 contained 14 acclimatized volunteers and 17 patients with HAPE. The microarray data GSE145935 was used to obtain the expression profile data of six groups of human astrocytes cultured for 24 h under normoxic conditions (5% CO_2_, 95% air) and hypoxic conditions (1% O_2_, 5% CO_2_, 94% N_2_) (Table 1). Finally, 1793 immune-related genes were identified using the ImmPort database (https://www.immport.org (accessed on 9 June 2022)).

The TCGA database (https://www.cancer.gov/tcga (accessed on 9 June 2022)), GTEx database (http://commonfund.nih.gov/GTEx/ (accessed on 9 June 2022)), and TIMER 2.0 database (http://timer.cistrome.org/ (accessed on 10 June 2022)) were mined for RNA sequencing, somatic mutation of the STC1 expression, as well as related clinical data between a variety of malignant tumors and healthy tissue types. The workflow is illustrated by Figure 12.

### 4.2. STC1 Expression Profiles

We used the HPA database, which can be found at https://www.proteinatlas.org/ (accessed on 9 June 2022), to evaluate STC1 expression in malignant as well as normal human cells. This allowed us to identify the mRNA and protein expression patterns that are present in human tissues. Analyzing the proteomes of genome-annotated TCGA tumor specimens was performed with the help of the CPTAC (https://proteomics.cancer.gov/programs/cptac (accessed on 10 June 2022)) database. The genetic modification of STC1 was investigated via the cBioPortal website (https://www.cbioportal.org/ (accessed on 10 June 2022)). We acquired eight datasets from the GEO database to verify the STC1 expression and its diagnostic value and confirm the findings of the pan-analysis (Table 1). The differentially expressed mRNAs (DE-mRNAs) and differentially expressed miRNAs (DE-miRNAs) were identified between two groups, respectively, with the criteria of log |(FC)| > 1 and *p* value < 0.05, using the R software (version 4.2.1) ‘limma’ package.

### 4.3. Identification of Target Genes of DE-miRNA

The target genes of DE-miRNA were predicted using three prediction databases: miRDB [36], miRTarbase [37], and TargetScan [38]. The genes predicted in all three databases were selected as the target genes of miRNA.

### 4.4. Protein–Protein Interaction (PPI) Network and Enrichment Analysis of STC1-Related Genes

The PPI network of STC1-binding proteins and the GO enrichment analysis were acquired from the STRING web (https://cn.string-db.org/ (accessed on 29 July 2023)). Then, we used Cytoscape (3.9.1) [39] to visualize the PPI network. On the basis of curated gene sets c2.cp.kegg. v7.0. symbols, GSEA was used to investigate KEGG as well as HALLMARK pathways of the STC1 gene, using the R software (version 4.2.1). The top three most significant pathways were shown.

### 4.5. Prognosis and Survival Analysis

Using forest plots and Kaplan–Meier curves, prognosis was correlated with STC1 expression in patients with 33 distinct malignancies. Researchers examined overall survival (OS), disease-specific survival (DPS), and progression-free interval (PFI). Using the Sangerbox tools (http://www.sangerbox.com/tool (accessed on 9 June 2022)), we were beneficial in detecting the hazard ratios (HRs) and the 95% confidence intervals (CI) [40].

### 4.6. Immune or Molecular Subtypes of Cancers and Clinical Phenotypes Analysis

Based on the TISIDB database [41], which describes tumor-immunity interactions by integrating research articles and multiple types of high-throughput data, we explored the correlation between STC1 expression and molecular and immune subtypes in multiple cancers. We downloaded the harmonized and standardized pan-cancer dataset from the UCSC (https://xenabrowser.net/ (accessed on 29 July 2023)) database and then calculated the differences in gene expression in each tumor in the samples with different clinical stages.

### 4.7. The Correlation of STC1 Expression with Cancer Cell Infiltration

In order to evaluate the infiltration levels of the six kinds of immune cells, comprising CD4+T cells, CD8+T cells, macrophages, B cells, neutrophils, and dendritic cells, in various malignancies, the TIMER 2.0 database was equipped. The scores for immunological and stromal components reflected their abundance. The total of prior ESTIMATE scores represented tumor purity indirectly [42]. For the presentation, we chose to highlight the three malignancies with the most significant immune infiltration data. Scatter plots were used to illustrate the relationships that were found between the levels of STC1 expression and immunological and stromal scores in various malignancies.

### 4.8. Single-Cell Sequencing Analysis of STC1

The TISCH (http://tisch1.comp-genomics.org/ (accessed on 30 July 2023)) database automatically parses and curates tumor single-cell RNA-seq datasets from GEO databases. The CancerSEA (http://biocc.hrbmu.edu.cn/CancerSEA/ (accessed on 30 July 2023)) is an analytic tool for studying cancer cell functions at the single-cell level, containing 14 tumor-related cellular functions of 900 cancer cells from 25 cancers.

### 4.9. Evaluation of Tumor Mutation Burden (TMB), and Tumor Microsatellite Instability (MSI), (Mis-Match Repair) MMR Gene Mutation and DNA Methylation of STC1

We downloaded the MSI scores and TMB of each tumor from the GDC (https://portal.gdc.cancer.gov/ (accessed on 9 June 2022)) after the MuTect2 software (version 4.1.0.0). The expression data for 37 cancer types were obtained by excluding those with less than three samples in a single cancer type. The TCGA database was searched to collect data concerned with STC1 gene expression levels in a variety of malignancies. Spearman’s correlation approach investigates the degree of association between expression levels of these MMR genes and STC1. CpGs that were lacking more than 10% of their values were taken out of the analysis. CpGs having an FDR < 0.05 and an absolute β difference > 0.2 were declared differentially methylated.

### 4.10. Quantitative RT-PCR (qRT-PCR)

The RAW264.7 cell line is a widely utilized mouse macrophage cell line appropriate for immunological studies, including the verification of immune gene functions. The RAW 264.7 cell lines were purchased from Guangzhou Cell cook Biotech Co., Ltd. (Guangzhou, China). Cells were cultured in culture media (Solarbio’s Dulbecco’s modified Eagle’s medium [DMEM] with high glucose). Hypoxia groups were cultured in a hypoxia chamber with 94% N_2_, 5% CO_2_, and 1% O_2_ for 18 h; control groups were cultured in an atmosphere of 5% CO_2_. The blood samples were collected from individuals diagnosed with HAPE and health control volunteers authorized by the XIZANG Center for Disease Control and Prevention Ethics Review Committee. Total RNA was extracted from the RAW 264.7 cell and blood of 20 HAPE patients and 20 control volunteers by using TRIzol reagent (Invitrogen, Carlsbad, CA, USA). RNA was reverse-transcribed into complementary DNA (cDNA) using StarScript II First-strand cDNA Synthesis Kit-II (GenStar, Beijing, China). qRT-PCR was performed by 2×RealStar Green Fast Mixture with ROX (GenStar, Beijing, China). The primers were synthesized by the Beijing Genomics Institute (BGI) (Beijing, China), as described in Appendix A. The specificity of these primer sequences was tested by the BLAST algorithm.

### 4.11. Statistical Analysis

Differences between the two groups were estimated using a two-tailed Student’s *t*-test and ANOVA was used to test the differences in the samples of multiple groups. All of the survival studies that were performed for this research made use of the log-rank test, the Kaplan–Meier curve, as well as the Cox proportional hazard regression model. Spearman’s test examined the correlation between the two variables. When *p* < 0.05, statistical significance was assumed.

## 5. Conclusions

Briefly, the discovery of STC1 has broadened our comprehension of the role of immune genes in the development of high-altitude diseases. Additionally, STC1 levels can act as independent predictive biomarkers to identify patients who will respond positively to immune checkpoint inhibitors. The integration of research on high-altitude diseases and pan-cancer research has fostered collaboration across various disciplines, including mountain medicine, molecular biology, and cancer research. This interdisciplinary collaboration can expedite the integration and innovation of knowledge, thereby advancing the fields of medicine and biological sciences. By integrating key genes related to high-altitude diseases with pan-cancer analysis, we can not only enhance our comprehension of the common biological processes across these conditions but also facilitate the development of novel treatment approaches, significantly enhancing the efficacy and efficiency of disease management.

## Figures and Tables

**Figure 1 ijms-25-09085-f001:**
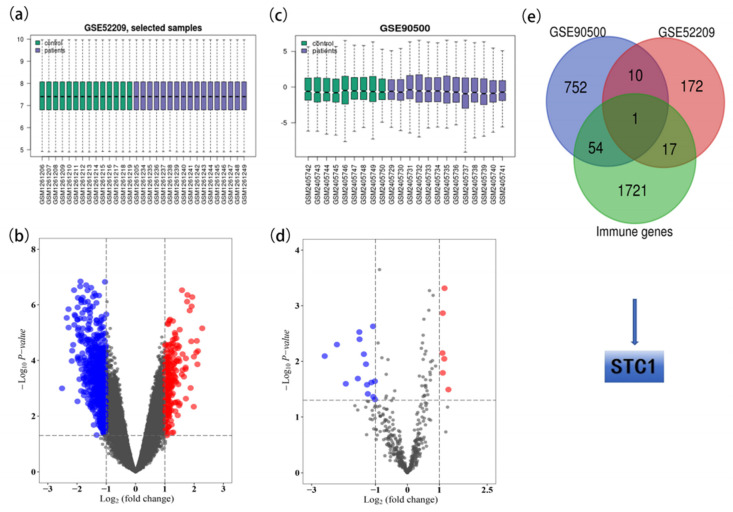
Identification of STC1 from GSE90500 and GSE52209. (**a**) The box diagram for sample quality control of GSE90500. (**b**) Volcano plot of DE-miRNAs in GSE90500. Red dots indicate up-regulated miRNAs, and blue dots indicate down-regulated miRNAs with *p* < 0.05 and |logFC| > 1. Gray dots indicate the genes that were not differentially expressed in AMS vs. non-AMS. (**c**) The box diagram for sample quality control of GSE52209. (**d**) Volcano plot of DEGs in GSE52209. Red dots indicate up-regulated mRNA, and blue dots indicate down-regulated mRNAs with *p* < 0.05 and |logFC| > 1. Gray dots indicate the genes that were not differentially expressed in control vs. HAPE. (**e**) Venn plot for identifying the STC1 gene.

**Figure 2 ijms-25-09085-f002:**
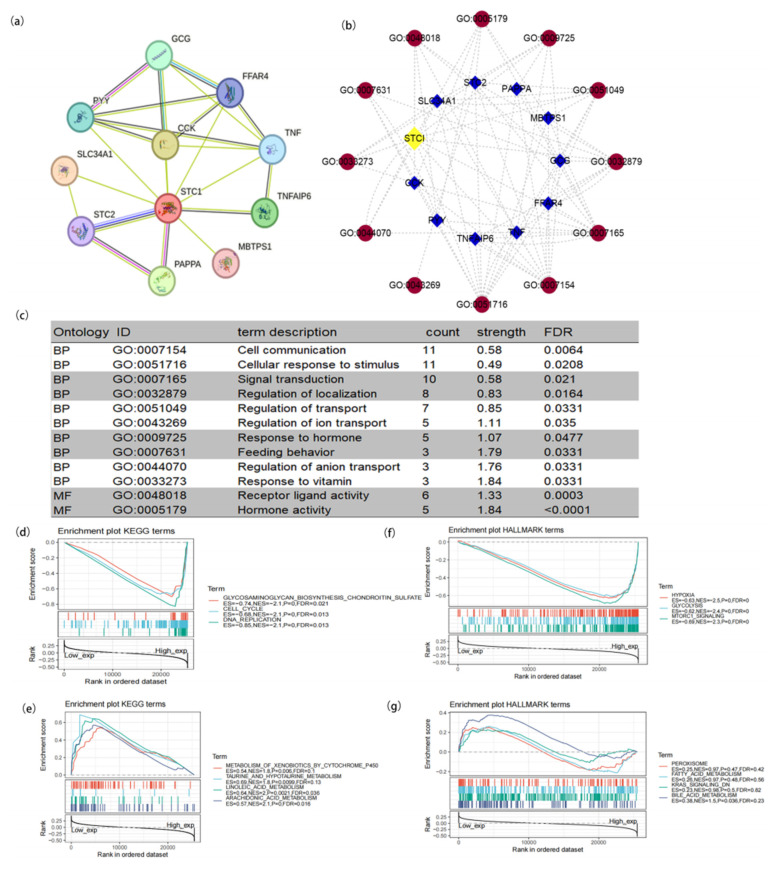
Protein–protein interaction (PPI) network and functional enrichment of STC1. (**a**) PPI network. (**b**) Visual network of GO analysis (blue: molecular; red: GO ID). (**c**) GO analysis of targeted binding proteins of STC1. (**d**–**g**) Enrichment analysis related to STC1 expression using GSEA. (**d**) High STC1 expression sample enriched KEGG gene sets. (e) Low STC1 expression sample enriched KEGG gene sets. (**f**) Samples with elevated STC1 expression enhanced the immunologic gene sets within the HALLMARK collection. (**g**) Samples with low STC1 expression enhanced the immunologic gene sets within the HALLMARK collection. Up-regulated genes are found on the left, close to the coordinates’ origin, while down-regulated genes are found on the right, away from the *x*-axis. Each line represents a specific gene set with its own color. The sole gene sets that were deemed significant have NOM *p* < 0.05 and FDR q < 0.05. In the plot, just a few leading gene sets were visible.

**Figure 3 ijms-25-09085-f003:**
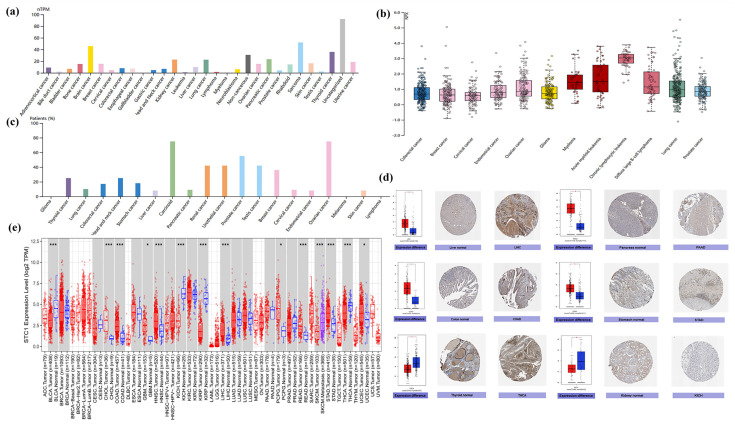
Differential expression of STC1 in tumor and normal tissues. (**a**) STC1 expression in tumor cell lines from the HPA database. (**b**) STC1 expression of relative plasma concentrations of proteins in blood from patients. (**c**) STC1 protein expression in cancer tissues with distinct extracellular staining. (**d**) STC1 gene expression in normal (**center**) and malignant (**right**) tissues, as shown in IHC pictures. (**e**) STC1 expression levels in various cancer types according to the TIMER by analyzing the TCGA database. The blue as well as red boxes indicate healthy and malignant tissues, respectively. (* *p* < 0.05, *** *p* < 0.001).

**Figure 4 ijms-25-09085-f004:**
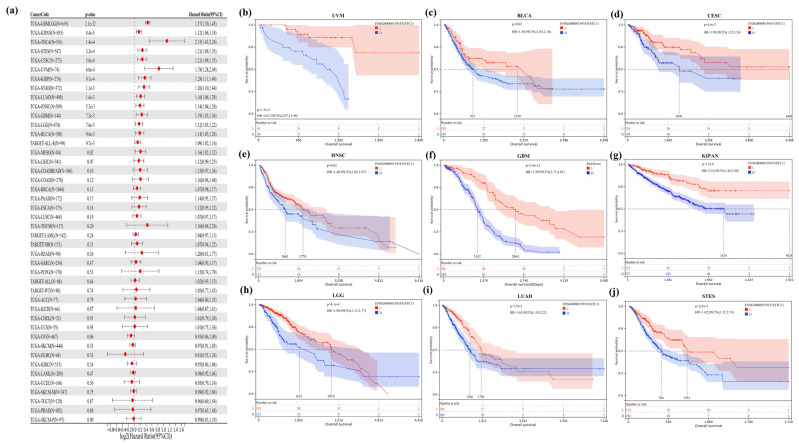
Correlation between STC1 gene expression and overall survival. (**a**) A forest plot of STC1 gene expression and patients’ OS in different cancers. (**b**–**j**) Kaplan–Meier curves for patients’ OS stratified by different expression levels of STC1 in UVM (uveal melanoma), BLCA (bladder urothelial carcinoma), CESC (cervical squamous cell carcinoma and endocervical adenocarcinoma), HNSC (head and neck squamous cell carcinoma), GBM (glioblastoma multiforme), KIPAN (pan-kidney cohort (KICH+KIRC+KIRP)), LGG (brain lower grade glioma), LUAD (lung adenocarcinoma), and STES (stomach and esophageal carcinoma).

**Figure 5 ijms-25-09085-f005:**
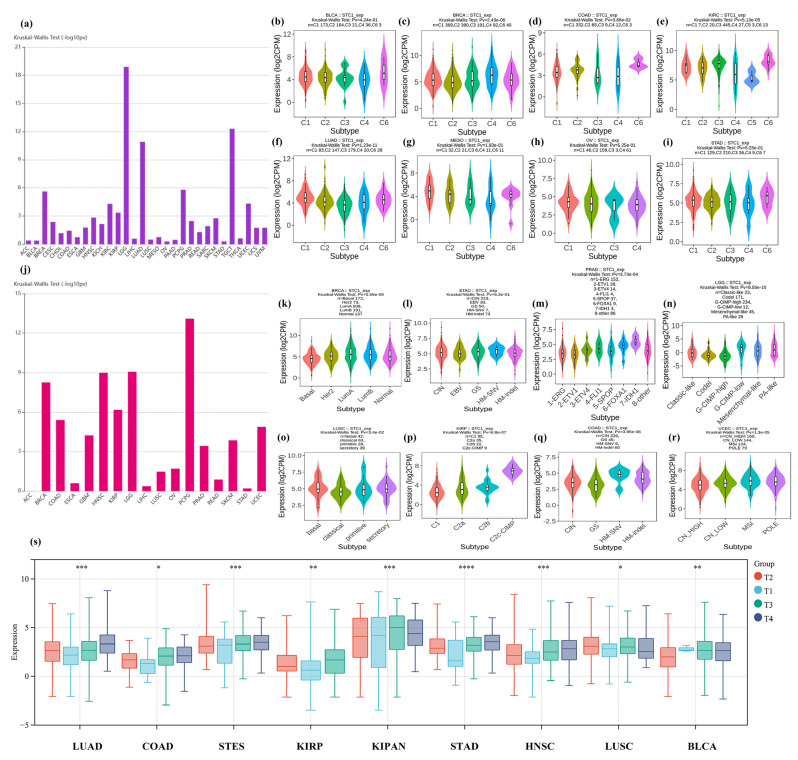
Correlation between STC1 expression and immune or molecular subtypes and clinical phenotypes across TCGA tumors. (**a**–**i**) immune subtypes. (**a**) Immune subtypes in 30 cancers. (**b**) BLCA (bladder urothelial carcinoma). (**c**) BRCA (breast invasive carcinoma). (**d**) COAD (colon adenocarcinoma). (**e**) KIRC (kidney renal clear cell carcinoma). (**f**) LUAD (lung adenocarcinoma). (**g**) MESO (mesothelioma). (**h**) OV (ovarian serous cystadenocarcinoma). (**i**) STAD (stomach adenocarcinoma). (**j**–**r**) Molecular subtypes. (**j**) Molecular subtypes in 17 cancers. (**k**) BRCA (breast invasive carcinoma). (**l**) STAD (stomach adenocarcinoma). (**m**) PRAD (prostate adenocarcinoma). (**n**) LGG (brain lower grade glioma). (**o**) LUSC (lung squamous cell carcinoma). (**p**) KIRP (kidney renal papillary cell carcinoma). (**q**) COAD (colon adenocarcinoma). (**r**) UCEC (uterine corpus endometrial carcinoma). C1: wound healing, C2: IFN-gamma dominant, C3: inflammatory, C4: lymphocyte depleted, C5: immunologically quiet, C6: TGF-b dominant. (**s**) Correlation between STC1 expression and clinical phenotypes. (* *p* < 0.05, ** *p* < 0.01, *** *p* < 0.001, **** *p* < 0.0001).

**Figure 6 ijms-25-09085-f006:**
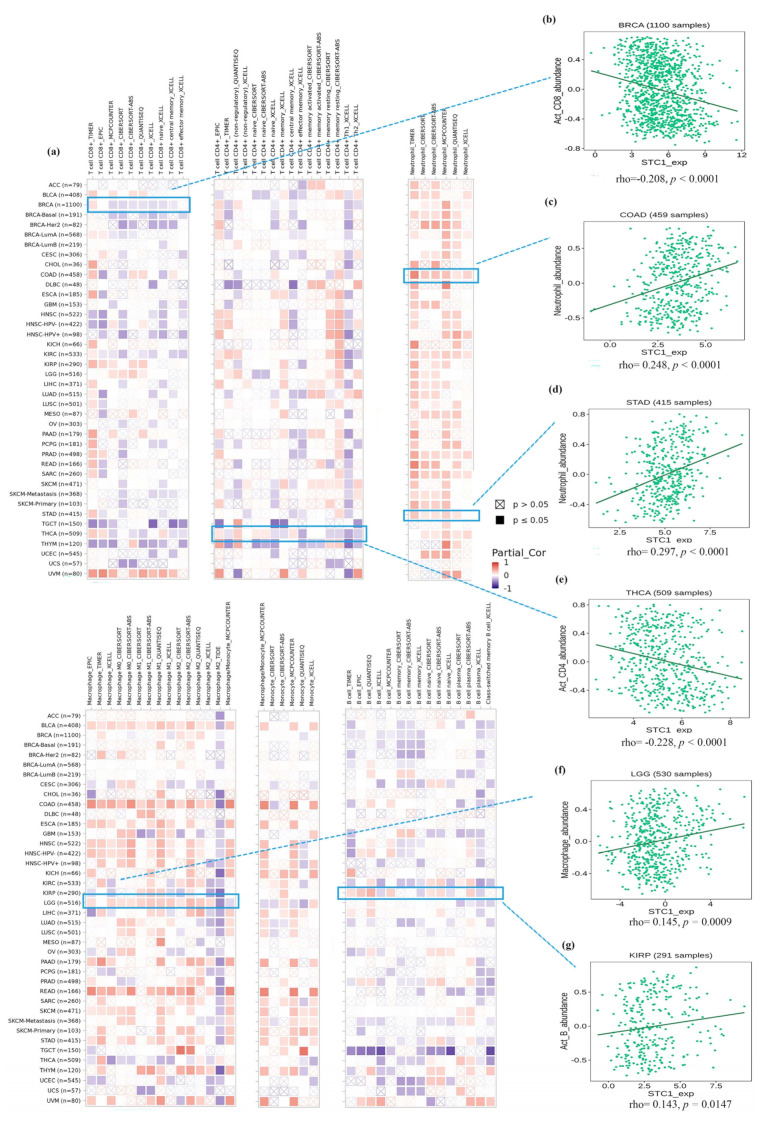
Immune cell infiltration analysis of STC1 in all malignancies within the TCGA. (**a**) Several algorithms were utilized to investigate the possible correlation between STC1 gene expression and the infiltration level of CD8+ T cell, CD4+ T cells, neutrophils, macrophages, monocytes, and B cells. (**b**–**g**) The correlation plot of those having significant r values based on the different algorithms. (**b**) BRCA. (**c**) COAD. (**d**) STAD. (**e**) THCA. (**f**) LGG. (**g**) KIRP.

**Figure 7 ijms-25-09085-f007:**
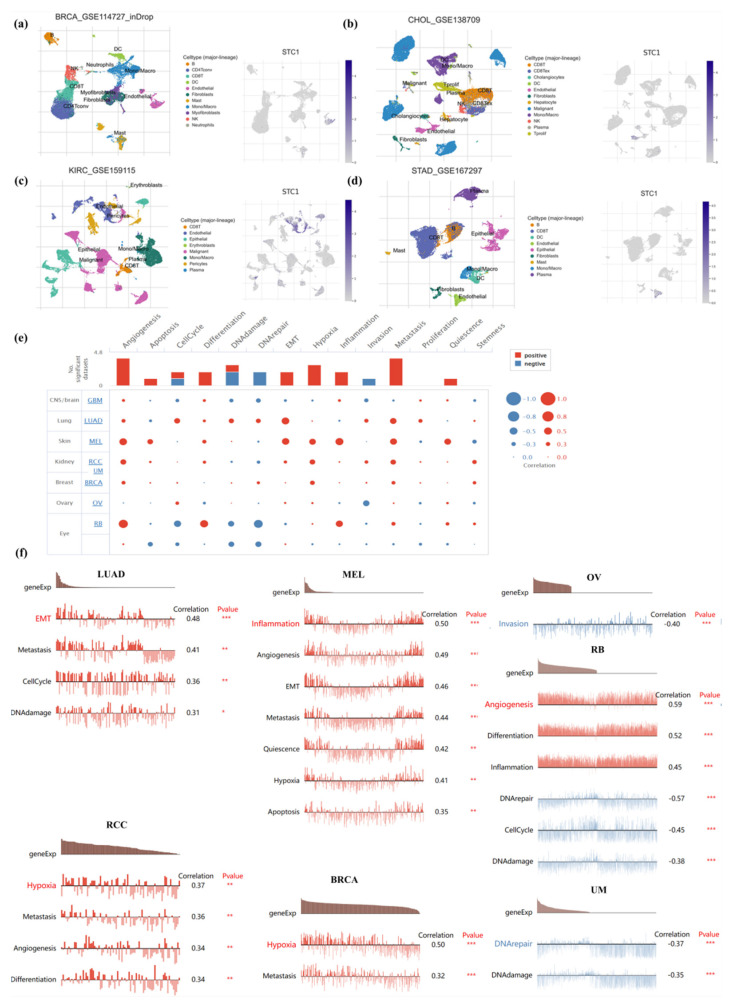
Single-cell analysis of STC1 in human cancers. (**a**–**d**) Composition and distribution of single cells and STC1 expression in single cells in BRCA, CHOL, KIRC, and STAD from the TISCH database. Each cell type is ranked by its average expression value. (**e**) Functional status of STC1 in different human cancers from the CancerSEA database. (**f**) Correlation analysis between functional status and STC1 in LUAD, RCC, MEL, BRCA, OV, RB, and UVM. (* *p* < 0.05, ** *p* < 0.01, *** *p* < 0.001).

**Figure 8 ijms-25-09085-f008:**
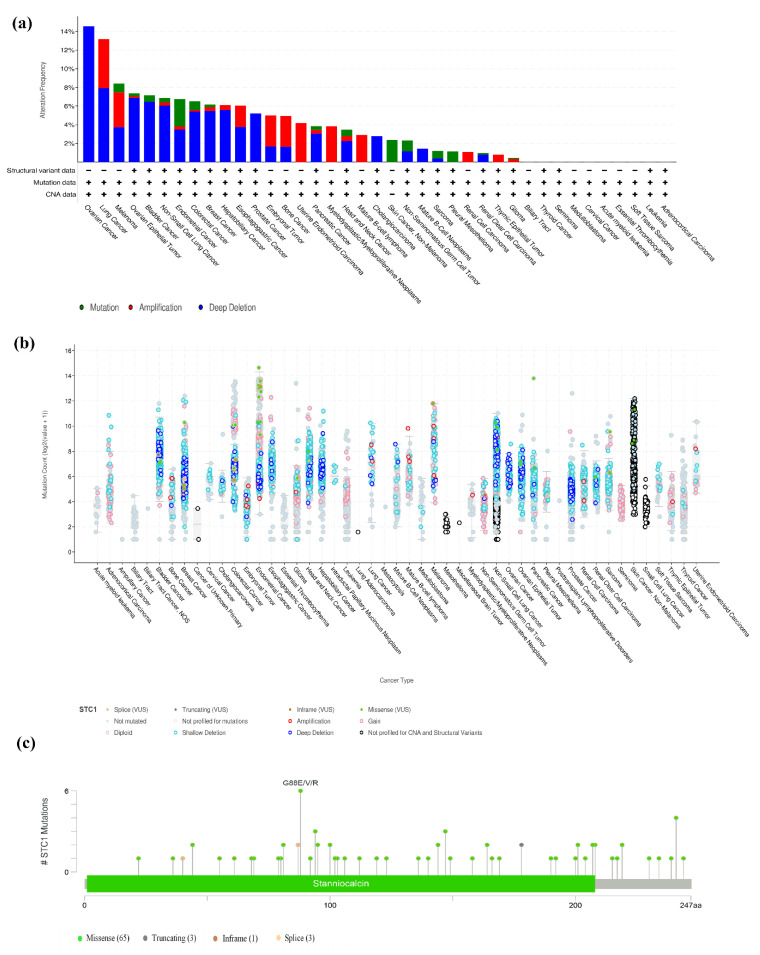
The nature of STC1 genetic mutations. (**a**) Genetic mutations in STC1 were often seen across numerous TCGA pan-cancer investigations, as shown by data from cBioPortal. (**b**) The cBioPortal database’s overall STC1 mutation count in distinct TCGA cancers. (**c**) The STC1 genetic mutation maps across protein domains in several malignancies.

**Figure 9 ijms-25-09085-f009:**
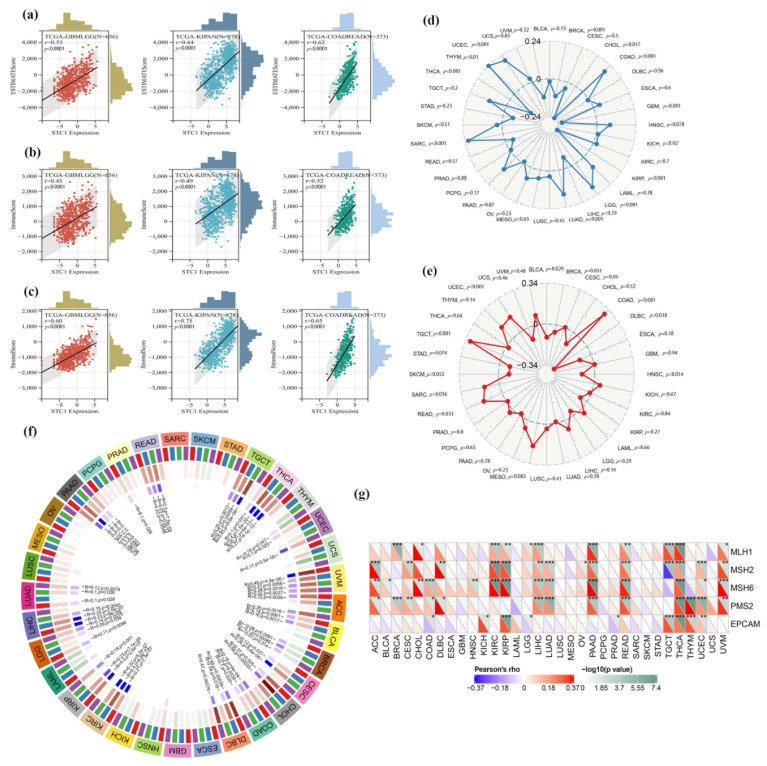
Relationship between TMB, MSI, MMR faults, methylation levels, and STC1 expression levels in distinct malignancies. (**a**–**c**) Top three cancers by ImmuneScore, StromalScore, and ESTIMATEScore, respectively. Correlation of STC1 mRNA expression and infiltration scores, calculated by TIMER from the TCGA database. (**d**,**e**) The correlations between STC1 expression as well as TMB and MSI in different malignancies. (**f**) STC1 expression and four methyltransferases are correlated (DNMT1: red, DNMT2: blue, DNMT3A: green, DNMT3B: purple). (**g**) STC1 expression correlates with essential genes implicated in five significant MMR genes. (* *p* < 0.05, ** *p* < 0.01, *** *p* < 0.001).

**Figure 10 ijms-25-09085-f010:**
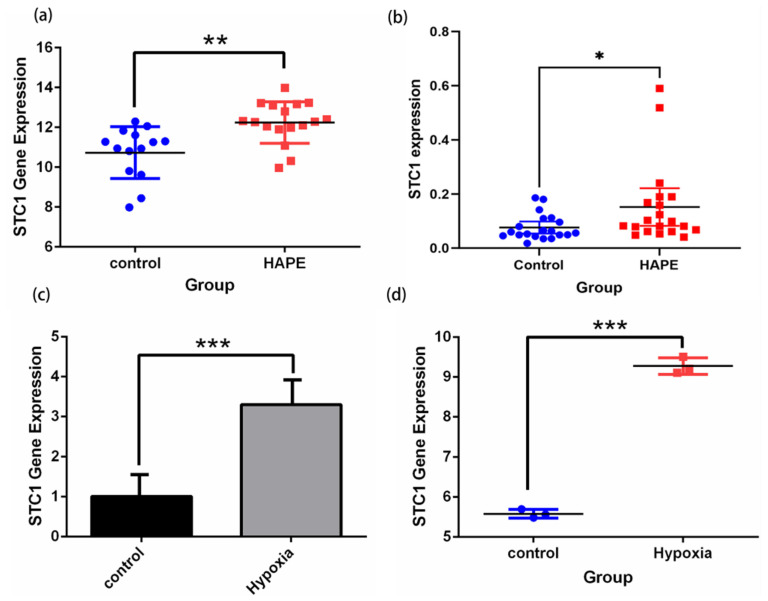
Validation of STC1 expression under hypoxia and normoxia by qRT-PCR tests and in GEO datasets. (**a**) STC1 expression of HAPE and control volunteers in GSE52209. (**b**) qRT-PCR confirmed STC1 expression differences in 20 pairs of HAPE and control volunteers. (**c**) STC1 expression in RAW 264.7 cells cultured in hypoxia rather than normoxia. (**d**) STC1 expression in GSE145935. (* *p* < 0.05, ** *p* < 0.01, *** *p* < 0.001).

**Figure 11 ijms-25-09085-f011:**
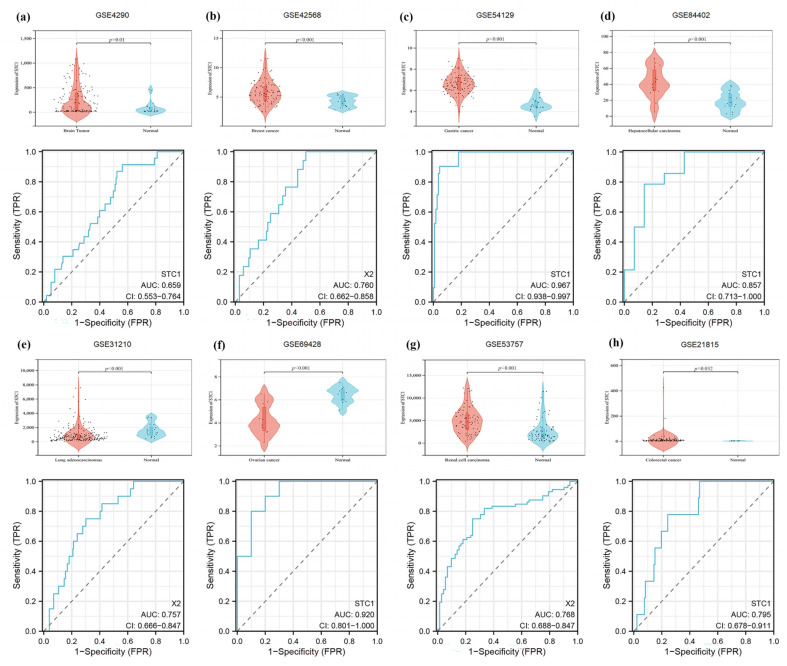
Validation of tumor-normal STC1 expression and its diagnostic value in GEO datasets. (**a**) Brain tumor in GSE4290. (**b**) Breast cancer in GSE42568. (**c**) Gastric cancer in GSE54129. (**d**) HCC in GSE84402. (**e**) Lung adenocarcinomas in GSE31210. (**f**) Ovarian cancer in GSE69428. (**g**) Renal cell carcinoma in GSE53757. (**h**) Colorectal cancer in GSE21815. The top graph is a violin plot of STC1 expression, and the bottom graph is the ROC curve of STC1.

**Figure 12 ijms-25-09085-f012:**
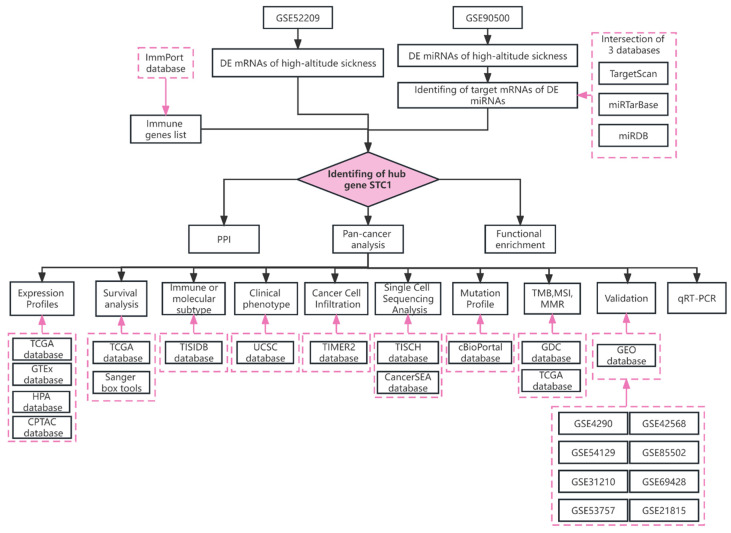
The workflow of the study.

**Table 1 ijms-25-09085-t001:** Dataset selection and application.

GEO Dataset	Application	Title	Platform
GSE90500	Test set	Circulating microRNAs as a signature predicting the occurrence of acute mountain sickness before exposure to high altitude	GPL18058 (Exiqon miRCURY LNA microRNA array, 7th generation)
GSE52209	Test set	Gene expression profiling under exposure to a high-altitude hypoxic environment	GPL9365 (Ocimum Biosolutions Human 40k OciChip)
GSE145935	Validation set	Transcriptomic analysis of human astrocytes in vitro reveals hypoxia-induced mitochondrial dysfunction, modulation of metabolism, and dysregulation of the immune response	GPL570 [HG-U133_Plus_2] Affymetrix Human Genome U133 Plus 2.0 Array
GSE4290	Validation set	Expression data for glioma samples from Henry Ford Hospital	GPL570 [HG-U133_Plus_2] Affymetrix Human Genome U133 Plus 2.0 Array
GSE42568	Validation set	Breast cancer gene expression analysis	GPL570 [HG-U133_Plus_2] Affymetrix Human Genome U133 Plus 2.0 Array
GSE54129	Validation set	Global gene expression analysis of gastric cancer by oligonucleotide microarrays	GPL570 [HG-U133_Plus_2] Affymetrix Human Genome U133 Plus 2.0 Array
GSE85502	Validation set	Genome-wide profiling of H3K27me3 in Drosophila primary spermatocytes	GPL15057NimbleGen Drosophila melanogaster Whole Genome 2.1M tiling array [DM_5_Catalog_tiling_HX1; DesignID 6725]
GSE31210	Validation set	Gene expression data for pathological stage I-II lung adenocarcinomas	GPL570 [HG-U133_Plus_2] Affymetrix Human Genome U133 Plus 2.0 Array
GSE69428	Validation set	Transformation of human fallopian tube stem cells and high-grade serous ovarian cancer	GPL570 [HG-U133_Plus_2] Affymetrix Human Genome U133 Plus 2.0 Array
GSE53757	Validation set	Gene array analysis of clear cell renal cell carcinoma tissue versus matched normal kidney tissue	GPL570 [HG-U133_Plus_2] Affymetrix Human Genome U133 Plus 2.0 Array
GSE21815	Validation set	Gene expression profiles in 132 laser microdissected colorectal cancer tissues	GPL 6480Agilent-014850 Whole Human Genome Microarray 4x44K G4112F (Probe Name version)

## Data Availability

The datasets analyzed during the current study are available in the TCGA database (https://www.cancer.gov/ccg/research/genome-sequencing/tcga (accessed on 9 June 2022)) and GEO database (https://www.ncbi.nlm.nih.gov/geo/ (accessed on 30 June 2023)).

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
