# Peer review of "Identification and Validation of STC1 Act as a Biomarker for High-Altitude Diseases and Its Pan-Cancer Analysis"

_ijms, 2024, doi:10.3390/ijms25169085_

Round 1

Reviewer 1 Report

Comments and Suggestions for Authors

The paper is interesting and well written. The authors investigated the role of STC1 gene in high-altitude diseases and explores its expression patterns in different types of cancer. I suggest to improve the discussion on pro-inflammatory cytokines underlining the role of Th17 cells, IL-31/IL-33 a is, vitamin D and microbiome in chronic inflammatory immune-mediated responses (see and add as references papers by Murdaca et al concerning these topics).

Comments on the Quality of English Language

Minor english editing

Author Response

Reviewer 1

You made many helpful comments and suggestions to improve the quality of our manuscript, and we thank you for them. We have studied comments carefully and have made correction which we hope meet with approval.

Comment 1:

I suggest to improve the discussion on pro-inflammatory cytokines underlining the role of Th17 cells, IL-31/IL-33 a is, vitamin D and microbiome in chronic inflammatory immune-mediated responses (see and add as references papers by Murdaca et al concerning these topics).

Response: Thank you for the suggestion. We did indeed overlook the discussion on the relationship between immune infiltrating cells and STC1, therefore, we have supplemented it, please refer to line 302-313.

Furthermore, our findings demonstrate a strong correlation between STC1 expression and the infiltration levels of immune cells, including CD8+ T cells, CD4+ T cells, neutrophils, macrophages, monocytes, and B cells. This suggests that STC1 may influence the tumor immune response by modulating the activation, migration, or functional dynamics of these immune cell populations. STC1 may influence the infiltration and activity of CD8+ T cells and CD4+ T cells by modulating pro-inflammatory or immunosuppressive factors, such as TGF-β, within the tumor microenvironment . STC1 may indirectly modulate the cytotoxic activity of CD8+ T cells through the regulation of IL-17 expression.STC1 may influence CD4+ T cells to secrete cytokines, including IL-17, IL-31, and IL-33, which modulate immune responses via autocrine or paracrine mechanisms, thereby either amplifying or suppressing immune activity within the tumor microenvironment. Similarly, STC1 may participate in tumor progression by affecting the polarization state of macrophages (M1 anti-tumor or M2 pro tumor). Additionally, STC1 may impact T cell persistence through its anti-apoptotic and pro-survival properties, thereby modulating their functional role within tumors.

Reviewer 2 Report

Comments and Suggestions for Authors

The study “Identification and Validation of STC1 act as a biomarker for high-altitude diseases and its Pan-Cancer Analysis” focuses on the role of the STC1 gene in high-altitude diseases and explores its expression patterns in different types of cancers.

The data presented in this study are comprehensive and valuable.

The entire manuscript is well written in all its parts, with an adequately described Materials and Method section and a properly explained and illustrated result section.

However, some minor obstacles must be corrected before the manuscript can be considered for publication.

These include:

3. Results

3.1. Identification of the DE-miRNAs and DE-mRNAs

The authors should explain which software was used to analyze the GSE90500 and GSE52209 data sets.

3.2. Identification of the key genes

The authors should additionally explain the number of genes presented in Figure 2. e.

3.3. PPI network and function enrichment of STC1

The authors have written: “To study the function of STC1, we used the STRING database to explore STC1-bind-208 ing proteins, GO terms, and conducted functional enrichment analysis for STC1-related  genes using GSEA.”  What software /version was used for GSEA analysis?

3.4. Analysis of STC1 expression between normal and tumor samples

The resolution of Figure 4 should be improved.

3.5. STC1 gene acted as a survival prognosis biomarker

The resolution of Figure 5 should be improved. Also, it would be helpful if the authors could provide the full names of cancer types analyzed in the figure legend.

3.6. STC1 expression in immune or molecular subtype of cancers and clinical phenotypes

it would be helpful if the authors could provide the full names of cancer types analyzed in the figure legend (Figure 6).

3.7. Immune cell infiltration analysis of STC1 in all malignancies

The resolution of Figure 7a (text) should be improved.

The same applies to Figure 9.

A minor revision of the manuscript is suggested.

Author Response

Reviewer 2

We thank you for the critical comments and helpful suggestions. We have taken all these comments and suggestions into account, and have made major corrections in this revised manuscript. The comments and suggestions are all valuable and very helpful for revising and improving our paper, as well as the important guiding significance to our researches.

Comment 1:

3.1. Identification of the DE-miRNAs and DE-mRNAs

The authors should explain which software was used to analyze the GSE90500 and GSE52209 data sets.

Response: Thank you for the reminding. In fact, we have already described the analysis software for differential genes in the 'Materials and Methods' section. Please refer to line 347-349.

The differentially expressed mRNAs (DE-mRNAs) and differentially expressed miRNAs (DE-miRNAs) were identified between two groups, respectively, with the criteria of log |(FC)| > 1 and P value < 0.05, using the R software ‘limma’ package.

Comment 2:

The authors should additionally explain the number of genes presented in Figure 2. e.

Response: Thank you for the suggestion.We have explained the number of genes in the three groups shown in Figure 2e. Please refer to line 81-84.

Common DE-mRNAs were identified as the overlapping genes between the 817 target genes of DE-miRNAs identified in GSE90500 and the 200 DE-mRNAs identified in GSE52209. In addition, STC1 was identified as the overlapping genes between the 1793 immune-related genes (Table S5) and 11 common DE-mRNAs (Figure 2e).

Comment 3:

3.3. PPI network and function enrichment of STC1

The authors have written: “To study the function of STC1, we used the STRING database to explore STC1-bind-208 ing proteins, GO terms, and conducted functional enrichment analysis for STC1-related  genes using GSEA.”  What software /version was used for GSEA analysis?

Response: We conducted GSEA analysis using R software (version 4.2.1) and added it to the 357th line of the manuscript.

GSEA was used to investigate KEGG as well as HALLMARK pathways of the STC1 gene, using the R software (version 4.2.1).

Comment 4:

3.4. Analysis of STC1 expression between normal and tumor samples

The resolution of Figure 4 should be improved.

3.5. STC1 gene acted as a survival prognosis biomarker

The resolution of Figure 5 should be improved. Also, it would be helpful if the authors could provide the full names of cancer types analyzed in the figure legend.

3.6. STC1 expression in immune or molecular subtype of cancers and clinical phenotypes

it would be helpful if the authors could provide the full names of cancer types analyzed in the figure legend (Figure 6). 

3.7. Immune cell infiltration analysis of STC1 in all malignancies

The resolution of Figure 7a (text) should be improved.

The same applies to Figure 9.

Response: Thank you for the reminding. We have remade Figure 4, Figure 5, Figure 7, and Figure 9, increasing their resolution. We also provided the full names of cancer types in the figure legend (Figure 5 and Figure 6). Please refer to line 599-602, line 605-612 .

 UVM (Uveal Melanoma), BLCA (Bladder Urothelial Carcinoma), CESC (Cervical squamous cell carcinoma and endocervical adenocarcinoma), HNSC (Head and Neck squamous cell carcinoma), GBM (Glioblastoma multiforme), KIPAN (Pan-kidney cohort (KICH+KIRC+KIRP)), LGG (Brain Lower Grade Glioma), LUAD (Lung adenocarcinoma), and STES (Stomach and Esophageal carcinoma).

(b) BLCA (Bladder Urothelial Carcinoma). (c) BRCA (Breast invasive carcinoma). (d) COAD (Colon adenocarcinoma). (e) KIRC (Kidney renal clear cell carcinoma). (f) LUAD (Lung adenocarcinoma). (g) MESO (Mesothelioma). (h) OV (Ovarian serous cystadenocarcinoma). (i) STAD (Stomach adenocarcinoma). (k) BRCA (Breast invasive carcinoma). (l) STAD (Stomach adenocarcinoma). (m) PRAD (Prostate adenocarcinoma). (n) LGG (Brain Lower Grade Glioma). (o) LUSC (Lung squamous cell carcinoma). (p) KIRP (Kidney renal papillary cell carcinoma). (q) COAD (Colon adenocarcinoma). (r) UCEC (Uterine Corpus Endometrial Carcinoma).
